# Design and Print Terahertz Metamaterials Based on Electrohydrodynamic Jet

**DOI:** 10.3390/mi14030659

**Published:** 2023-03-15

**Authors:** Tong Yang, Xinyu Li, Bo Yu, Cheng Gong

**Affiliations:** Institute of Modern Optics, Nankai University, Tianjin Key Laboratory of Micro-Scale Optical Information Science and Technology, Tianjin 300350, China

**Keywords:** metamaterials, terahertz, electrohydrodynamic jet printing, step-motor driven

## Abstract

Terahertz metamaterials are some of the core components of the new generation of high-frequency optoelectronic devices, which have excellent properties that natural materials do not have. The unit structures are generally much smaller than the wavelength, so preparation is mainly based on semiconductor processes, such as coating, photolithography and etching. Although the processing resolution is high, it is also limited by complex processing, long cycles, and high cost. In this paper, a design method for dual-band terahertz metamaterials and a simple, rapid, low-cost metamaterial preparation scheme based on step-motor-driven electrohydrodynamic jet technology are proposed. By transforming an open-source 3D printer, the metamaterial structures can be directly printed without complex semiconductor processes. To verify effectiveness, the sample was directly printed using nano conductive silver paste as consumable material. Then, a fiber-based multi-mode terahertz time-domain spectroscopy system was built for testing. The experimental results were in good agreement with the theoretical simulation.

## 1. Introduction

Metamaterial is a kind of artificial composite subwavelength structural material with excellent performance, which can surpass that of natural materials. Its performance mainly comes from the precise geometric configuration and size. By changing the structural shape, size, composition materials and other parameters of the unit, it can achieve effective regulation of electromagnetic parameters [1], so as to achieve features that traditional materials do not have or cannot achieve, such as a negative refractive index [2,3], super lens [4], electromagnetic stealth [5], perfect absorption [6,7,8,9], hyperbolic dispersion [10,11] and so on. In 2008, N.I. Landy et al. proposed a perfect metamaterial absorber working in the microwave band [7]. The absorber is composed of three layers: the top layer is a metal open resonance ring, the middle layer is a dielectric layer, and the bottom layer is a metal strip. The measured results showed that the metamaterial absorber had an absorption peak greater than 88% at 11.5 GHz. Since then, metamaterial-based absorbers have aroused great interest and gradually become a research hotspot. Through continuous improvement, metamaterial absorbers have developed from microwave band [12,13] to terahertz [14,15], near-infrared [16], mid-infrared [17,18], long-infrared [19], and even visible light band [20,21]. Their absorption characteristics have also increased from the initial single band [7,14] to multi-band [22,23] and wide band [24,25,26]. For example, in 2018, Luo’s team proposed an efficient ultra-wideband absorber for ultraviolet to near-infrared, which showed an average absorption of more than 95% in the wavelength range of 200~900 nm [26]. In 2018, Meng et al. proposed a multi-frequency terahertz absorber with a periodic square metal structure with T-shaped gaps, which had four absorption peaks, and absorption of above 98% [23]. In 2021, Fu et al. designed a three narrowband terahertz absorber based on a symmetrical double-trident structure and obtained 99.68%, 98.19% and 99.94% absorptivity at 0.909 THz, 1.899 THz and 2.395 THz, respectively [27]. In 2022, Khatami et al. realized a four-band terahertz perfect classical absorber based on the overlapping combination structure of two rectangles and circles; the average absorptivity of the four absorption peaks was 98.22% [28]. It can be seen that most of the metamaterials are fabricated by photolithography and etching processes based on semiconductor processing, including laser direct writing, ultraviolet lithography, reactive ion etching, deep silicon etching, electron beam etching and other processes. For example, in 2008, Landy et al.’s first metamaterial absorber was fabricated by photolithography, with a minimum size of 0.6 mm [7]. In 2017, Le et al. used electron beam etching technology to prepare gold nano resonant structures on silicon oxide layers, and, finally, obtained infrared-band metamaterial absorbers [29]. In 2019, Zhao et al. prepared a terahertz wave band metamaterial absorber using photolithography and deep reactive ion etching [30].

However, the traditional coating, photolithography and etching processes are complex, time-consuming, and costly. In addition, some preparation processes require the use of toxic and corrosive chemical reagents, which can easily pollute the environment. Therefore, it is of important research significance to seek a simple, rapid and low-cost processing method. Electrohydrodynamic jet printing, also known as E-jet printing, is a micro-droplet spray deposition molding technology based on electro-hydrodynamic (EHD) principles developed by Park and Rogers et al. in 2007 [31]. Different from the traditional micro-nano machining process, electrohydrodynamic jet printing can realize the direct fabrication of metamaterial resonant structures [32,33,34] without photolithography. It is also different from the “push” method of the traditional jet printing process. It uses an electric field to drive and “pull” the tip of the Taylor cone to generate an extremely thin jet, which ejects tiny droplets much smaller than the size of the nozzle to deposit on the substrate. Therefore, electrohydrodynamic jet printing has the advantages of high resolution [35], good compatibility, simple process, etc. In addition, the forms of consumables for electrohydrodynamic jet printing are flexible and diverse [36,37], including solutions, suspensions and high-viscosity materials. As a revolutionary technology, it could be used to replace traditional lithography and etching technology to realize the large-scale manufacture of micro-nano devices [37].

The electrohydrodynamic jet printing system is usually composed of an air pump, an air pressure valve, a nozzle, a high-pressure source, a mobile platform and a computer control system. The air pump provides pressure to push the printing consumables into the nozzle and the thrust is controlled by the air valve. However, pneumatic devices, such as air pumps occupy a large volume and have high costs, which makes it impossible to achieve low cost and miniaturization of devices. Therefore, we propose a metamaterial preparation scheme based on step-motor-driven electrohydrodynamic jet printing technology. The high-precision stepper motor is used to drive printing consumables so as to replace the pneumatic drive, which can accurately control the printing speed and the feeding amount of consumable materials and is small and low in cost. Nano conductive silver paste is used as a printing consumable to directly print metamaterial structures without complex processes, such as photolithography and etching. In addition, to improve the flexibility and practicability of terahertz devices, we proposed a design method of dual-band terahertz metamaterials, which has the characteristics of polarization insensitivity and wide incident angle. Next, a sample was directly printed using step-motor-driven electrohydrodynamic jet printing technology. Finally, we built a fiber-based multi-mode terahertz time-domain spectral system for testing. The highest measured absorption rate was greater than 90%, which represents good consistency between theory and experiment.

## 2. System Design and Principle

Figure 1 shows the principle of the metamaterial preparation system based on step-motor-driven electrohydrodynamic jet technology. Figure 1a displays the E-jet printing head composed of a step-drive motor, a screw, a storage syringe and a support frame. Figure 1b shows how the print head is installed on an open-source 3D printer (Ultimaker 2+) through modification; its X-Y-Z axis movement accuracy can reach 12.5 μm. Figure 1c is a schematic diagram of the Taylor cone generated by applying a high-voltage electric field.

The principles of operation of an E-jet print head can be described as follows: connect the screw slide with the piston of the storage syringe, drive the screw to rotate through a high-precision stepping motor, and then control the slide to move up and down; by controlling the rotation speed of the stepper motor, the up and down moving speed of the slider can be controlled, so as to achieve high-precision control of the injection feeding speed of the electrohydrodynamic jet printing. According to the minimum pulse speed of the stepping motor, the minimum moving speed of the slider is calculated as 1.95 μm/s, and the capacity of the matching syringe is 1 mL; then, the minimum feeding flow of the overall device can be calculated as 33.6 nL/s.

The 3D printing consumables we use are conductive silver paste with high viscosity and high conductivity. The nozzle matched with the syringe is a 27 G stainless steel needle with an inner diameter of 0.19 mm. According to Figure 1b, the E-jet print head is installed on the 3D mobile slide of the 3D printer. By inputting the designed structural model into the 3D slicing software, the computer can control the 3D printer to directly print the required metamaterial structure after slicing. As shown in Figure 1c, we add a high-voltage power supply, which can provide 0~30 kV high-voltage. The positive pole of the power supply is directly connected to the stainless-steel nozzle and the printing platform is grounded, thus realizing the hardware construction of the electrohydrodynamic jet printing system.

According to the principles of electrohydrodynamic jet technology, when the nozzle is connected to a positive high voltage and the substrate is grounded, an electric field is formed between the nozzle and the substrate, and the suspended droplets deposited at the nozzle are energized to generate induced charges and collect on the conical liquid surface. These droplets are subjected to normal and tangential electric field forces in the electric field, as well as their own gravity, the downward pressure applied by the motor to the droplet, and the surface tension and viscosity force of the droplet. With increase in the electric field, the tangential electric field force of the liquid surface stretches the liquid surface to form a conical liquid surface. When the above forces are balanced, the corresponding conical liquid surface is called a Taylor cone [37]. At this time, the electric field forces on the Taylor cone can be expressed as [38,39]:(1)$$F=qE-\frac{1}{2}\nabla E^{2}\varepsilon_{n}+\frac{1}{2}\nabla E^{2}\rho (\frac{\partial \varepsilon_{n}}{\partial \rho })$$

Here, *q* is the free-charge quantity, *ε_n_* stands for the permittivity of the printing material, *E* is the applied electric field intensity, and *ρ* stands for the density of the suspension droplets. The first term on the right side of Equation (1) is the Coulomb force, the second term is the polarization stress, and the third term is the electrostriction force, which can be ignored because the fluid is incompressible. During the formation of the Taylor cone, the flow of the droplet is affected by the balance between the viscous drag in the nozzle and the polarized electric stress at the liquid/air interface. The drop deployment rate can be estimated as *Q* in terms of the Poiseuille-type relation [40].
(2)$$Q\approx \frac{\pi R^{4}}{128\mu L}(\frac{\varepsilon_{n}E^{2}}{2}-\frac{2\sigma }{R}+\Delta P)$$
where *R* is the radius of the stainless-steel nozzle, *μ* is the viscosity of the liquid, *L* is the length of the nozzle, *σ* is the surface tension coefficient of the suspension droplets, and Δ*P* is the hydrostatic pressure with respect to the nozzle.

By continuously increasing the voltage, the force at the tip of the Taylor cone is unbalanced. The whole cone becomes longer and thinner under the combined action of the electric field force and surface tension, and, finally, forms a continuous jet or pulsating droplet at the tip. At this moment, the applied voltage is the minimum critical voltage *V_E_*, which satisfies the following equation [41]:(3)$$V_{E}{}^{2}=(0.117\pi R\sigma )(\frac{4H^{2}}{L^{2}})[\ln(\frac{2L}{R})-1.5]$$

Here, *H* is the distance from the stainless-steel nozzle to the grounding base plate.

When a stable cone jet is realized, for “high-conductivity” liquid (≥10^−5^ S/m), the jet flow rate can be represented by the characteristic flow rate *Q_m_* [40,42].
(4)$$Q_{m}∼\frac{\sigma \varepsilon_{n}}{\kappa \rho }$$
where, *κ* is the conductivity and the subscript *m* is a scaling variable. To maintain a stable jet flow, the supply flow must be balanced with the jet flow, which also means that the supply flow is related to the parameters, including *σ*, *ε_n_*, *κ* and *ρ*.

At this time, the voltage determines the quality of the conical jet. When the applied voltage is less than *V*_E_, the electric field force cannot break the surface tension and the conical jet mode will not appear. When the applied voltage is far greater than *V*_E_, the multi-cone jet mode will appear at this time, the jet direction will tilt, the number of jet strands will be difficult to control, and high-precision printing cannot be achieved. Therefore, when the external applied voltage is close to *V*_E_, a better single-cone jet mode can be formed, and high-precision printing can be achieved.

The above analysis shows that the conditions for achieving stable conical jet printing are mainly related to the feed flow rate and the electric field generated by the externally applied voltage. However, according to research on the theory of electrohydrodynamic jet printing technology published in the literature [43,44,45,46], the feed flow rate and the applied electric field are related to the external conditions, such as the nozzle diameter, the distance between the nozzle and the grounding substrate, the nozzle length, as well as the dielectric constant, conductivity, surface tension coefficient, viscosity coefficient, density and other parameters of printing consumables. In addition, achieving stable high-precision printing is also related to the moving speed of the E-jet printing head. Therefore, to achieve stable conical jet printing in the actual experimental process, it is necessary to fully consider the above factors and explore the appropriate printing conditions.

First, we selected a 27 G stainless steel nozzle (nozzle length: 13 mm, inner diameter: 0.19 mm), and then determined a kind of high viscosity and high conductivity printing consumable. Since the printing distance directly affects the electric field force on the Taylor cone liquid surface, the distance between the tip nozzle and the grounding substrate was fixed at 0.3 mm. Through simplifying the experimental conditions and preliminary optimization, we focused on three printing parameters that need to be adjusted experimentally: the feed velocity, the applied voltage and the printing speed.

We tested the electrohydrodynamic jet printing effect under different conditions and characterized the printed lines as shown in Figure 2. The printing material used was conductive silver paste with high conductivity and high viscosity; its conductivity was 5 × 10^6^ S/m, the silver content was 70–80%, and the viscosity was as high as 20,000 mPa∙s (25 °C). Figure 2a is a schematic diagram of printing by applied voltage to form a conical jet. Figure 2b shows five printing effects under different printing parameters. The specific three parameters are the applied voltage, feeding speed and printing speed, as shown in the subscript.

It can be seen that the system can achieve stable E-jet printing in the voltage range of 2.4~2.6 kV. We found in the experiment that, when the applied voltage was too low (<2.4 kV), the induced electric field near the nozzle tip was weak, and the printing material at the Taylor cone could not maintain a continuous conical jet, leading to a breakpoint in the printing silver line. With increase in the applied voltage, such as by 2.4 kV, the intensity of the induced charge near the nozzle tip gradually increased, and the Taylor cone was able to maintain a continuous jet. However, when the voltage was too high (>2.6 kV), the excessive electric field strength caused jet offset, leading to bending of the printing silver line. Therefore, stable E-jet printing was achieved using an applied voltage in the range of 2.4~2.6 kV during the experiment.

The feed speed and printing speed can be adjusted in coordination with each other. Generally, when the feeding speed is high and the printing speed is low, the printing line width is wide. When the feeding speed is low and the printing speed is high, the printing line width is narrow. It should be noted that, when the feeding speed is high and the printing speed is low, the printing line width is wide and the printing efficiency is low, which does not offer the advantages of large area rapid preparation of electrohydrodynamic jet printing. If the printing speed is increased, the line width can be reduced by dragging the conical jet. However, if the printing speed is too fast, it will not match the feeding speed, resulting in discontinuous printing lines. Therefore, the appropriate printing speed is 10~20 mm/s, and the printing line width can be further reduced by further reducing the feeding speed. At present, the minimum line width can reach 68 μm. However, when the line width is reduced to 68 μm, the printed silver line is affected by the moving stability of the print head, which may cause the problem of printing silver line bending. Therefore, in this system, the minimum line width that is more uniform and has good morphology is 105 μm, which can be used for printing metamaterial structures.

## 3. Metamaterial Design Method and Simulation

In recent years, metamaterials have become an important research focus due to their unique advantages and a variety of basic structures have been developed, among which the sandwich structure based on “metal-dielectric-metal” is one of the most classic structures. In the field of electromagnetic absorbers, many metamaterial absorbers use the sandwich structure. The top layer of the structure is a sub-wavelength resonant pattern, the middle layer is a dielectric layer with a certain thickness, and the bottom layer is a metal or conductive layer. However, according to the literature on metamaterials, there are many kinds of resonance patterns on the top layer, which makes it difficult to find rules and design methods. Therefore, we summarize and propose a simple metamaterial design method based on sandwich structure, which can achieve polarization insensitive and wide-incidence-angle dual-band absorption.

Figure 3 shows a schematic diagram of the design method. The first step is to determine the frame of the top resonant pattern. The frame is composed of four symmetrical metal diagonal lines at four corners, as shown in Figure 3a. The second step is to select the central structure shape of the top resonance pattern. The shape should be symmetrical to realize the characteristics of polarization insensitivity, as shown in Figure 3b. The third step is to determine the initial structural parameters of the resonant pattern (such as the line width, radius, material, and so on), using commercial electromagnetic simulation software to simulate and obtain the absorption spectrum, as shown in Figure 3c,d. The fourth step is to optimize the parameters according to the simulation results to determine whether the required absorption function is achieved. If it is not reached, go back to the second step to readjust the shape and parameters. If it is reached, go to the next step. Finally, according to the simulation and optimization results, the digital 3D model file of the structure, as shown in Figure 3e, is obtained.

Based on the above design method, we designed a dual-band metamaterial absorber, as shown in Figure 4. Its dielectric layer is selected as sodium-calcium glass, which also serves as a support layer. The top-layer resonant pattern is hexagonal, and its material is conductive silver wire. It can be directly printed on the surface of the dielectric layer by the electrohydrodynamic jet printing method. Then, the whole conductive silver film can also be printed on the bottom of the sodium-calcium glass using a similar printing method so that it covers the whole plane, acts as the bottom layer of the metamaterial absorber, blocks the transmission of terahertz wave, and forms a magnetic resonance with the top layer structure. Through optimization, the final structure and size parameters of the absorber are shown in Figure 4. Its periodic unit is a hexagonal grid, the center is a hexagon formed by silver lines, the four corners are connected by silver lines, the width of the silver lines is W_1_ = 120 μm, the length of the inner ring of the hexagon is L = 200 μm, and the period of the unit is *p* = 1120 μm. The thickness of the dielectric layer glass is t_2_ = 0.7 mm, the thickness of the top conductive silver wire t_1_ = 2.8 μm, and the thickness of the bottom conductive silver paste film t_3_ = 2.8 μm.

Because the unit structure size is sub-wavelength, the metamaterial absorber can be regarded as an effective medium and characterized by a complex permittivity *ε_eff_* and complex permeability *μ_eff_*. The top and bottom resonant structures and dielectric layers are strongly coupled with electric and magnetic fields to achieve impedance matching. The equivalent impedance *z* = (*μ_eff_*/*ε_eff_*)^1/2^ is consistent with the free-space impedance, thus minimizing reflection. The conductive silver film at the bottom layer can block the transmission of terahertz, thus minimizing the transmission. After simultaneously reducing reflection and transmission, the absorption can be maximized, or even perfectly absorbed.

In general, due to the complex shape of metamaterial resonant structures, it is difficult to accurately calculate the reflection and transmission characteristics through analytical methods. At present, the most common method is to use a numerical calculation method to solve the problem through 3D electromagnetic simulation software. The absorber’s reflectance and transmission can be acquired by simulating the complex frequency-dependent S parameters, *S*_11_ and *S*_21_. Therefore, we model metamaterial structures through the commercial 3D electromagnetic simulator CST Studio Suite 2016. The periodic boundary conditions are adopted and solved using a frequency domain solver, with a frequency range of 0.18~0.35 THz. We simulate the case of electromagnetic waves vertically incident on the top surface of THz metamaterial absorber. We set the THz wave to enter from the top structure of the absorber and exit from the bottom, and then calculate the corresponding S parameters. Then, the absorptivity *A* can be calculated by
(5)$$A=1-R-T=1-{\left|S_{11}\right|}^{2}-{\left|S_{21}\right|}^{2}$$
where *R* = |*S*_11_|^2^ and *T* = |*S*_21_|^2^ are the reflectance and transmission, respectively. In our design the transmission *T* is zero because of the printed conductive silver film ground. Therefore, the absorptivity can be given by *A* = 1 − *R*. The absorptivity curve of the metamaterial absorber is shown in Figure 5; there are two main absorption peaks, *A*_1_ = 97% and *A*_2_ = 98%, corresponding to two resonance frequencies 0.228 THz and 0.311 THz, respectively.

The physical origin and explanation of the absorptivity are as follows: the diagonal frame and central hexagon of the metamaterial cell can be regarded as two kinds of electric resonators, which can be strongly coupled with the electric field of the incident electromagnetic wave at 0.228 THz and 0.311 THz, as shown in Figure 5a and Figure 5b, respectively. At the same time, by adding a conductive ground plane, the plane and the electric resonators can be coupled with the magnetic components of the incident electromagnetic wave and induce resonance currents. The electric and magnetic responses can then be tuned independently. The electrical response can be adjusted by changing the geometric shape and unit size of the electric resonators; the magnetic response can be adjusted by changing the distance between the bottom layer and the electric resonators and the material parameters of the intermediate dielectric layer. The effective permittivity and permeability of metamaterials can be regulated by the regulation of electrical and magnetic responses; thus, the equivalent impedance can be regulated. When the equivalent impedance is matched with the free-space impedance, the perfect absorption of the dual-band can be achieved.

To provide a further interpretation, the transmission line model of the metamaterial cell is described in Figure 5c. *R_m_* stands for the electromagnetic loss of the dielectric layer. The two electric resonators (the diagonal frame and the central hexagon) are equivalent to two parallel RLC circuits. *R*_1_, *R*_2_ represent the ohm resistance of the electric resonators, *L*_1_, *L*_2_ are the equivalent inductance of the electric resonators, and *C*_1_, *C*_2_ are the equivalent capacitance of the resonators, respectively. According to the RLC circuit model, the resistance will not affect the resonance frequency, so we omitted it for simplicity. The inductance and capacitance can approximately describe the resonance of the metamaterial structures. Then the resonance frequency of the metamaterials can be given by:(6)$$f_{i}∼\frac{1}{2\pi \sqrt{L_{i}C_{i}}}\begin{array}{cc} & i=1,2. \end{array}$$

Here *i* represent the parameters of two electric resonators. Equation (6) demonstrates that the resonance frequency is an approximate linear function of the (*LC*)^1/2^. Therefore, the two resonators correspond to two resonant frequencies, producing dual-band absorption.

Furthermore, in practical applications, electromagnetic waves have different polarization states and incident states. The absorber should preferably be insensitive to polarization and the incident angle. Therefore, we analyze the absorption characteristics of the absorber at different polarization angles and different incidence angles. Figure 6a shows the simulation results of the absorption spectrum at different polarization angles. The absorption spectrum is almost constant and is not affected by the change in the polarization angle when the polarization angle is varied from 0 to 90°. Because its structure has central symmetry, it is insensitive to the polarization angle. Figure 6b shows the simulation results of the absorption spectrum at different incident angles. The absorption bandwidth becomes narrow as the incident angle increases from 0° to 50°. However, the absorptivity of the two peaks remains above 90%, which indicates that the designed dual-band absorber has good polarization insensitivity and wide incidence angle characteristics.

## 4. Printing and Measurement

After the simulation and optimization, we begin to process the THz metamaterial absorber. Figure 7 illustrates the process flow of the electrohydrodynamic jet printing, which starts with the spray-coating of conductive silver film (thickness is 2.8 μm) onto a sodium calcium glass substrate (thickness is 700 μm). The specific printing process is as follows: Firstly, a piece of soda-calcium glass of 55 mm × 45 mm × 0.7 mm is selected as the dielectric layer of the metamaterial, and the electrohydrodynamic jet printing system is used to make the conductive layer on the lower surface of the glass by spraying process. This spraying process does not require high-precision structure printing, so the printing parameters are slightly different, as follows: the printing distance is increased to 10 mm, the corresponding applied voltage is also increased to 3.5 kV, the feeding speed is 3.97 μm/s, the E-jet print head is fixed right above the glass, the conductive consumables are still conductive silver paste with high viscosity and high conductivity, and the 27 G stainless-steel nozzle is also selected, as shown in Figure 7I. After starting the feeding and spray printing, the conductive silver paste forms a Taylor cone at the nozzle and produces a short conical jet, which then falls down in the form of a spray, forming a layer of conductive silver paste coating on the glass, with the layer gradually thickening with time, as shown in Figure 7II. Then, we dry the sprayed conductive silver paste layer with a hot air gun at 50 °C, and finally obtain the lower surface of the metamaterial, as shown in Figure 7III. Next, the metamaterial subwavelength resonance structure is prepared on the upper surface of the glass, as shown in Figure 7IV,V. The production process requires high-precision electrohydrodynamic jet printing, and the printing parameters are fine-tuned according to the three parameters shown in Figure 2: the feed flow rate (1.98 μm/s), the applied voltage (2.4~2.6 kV) and the printing speed (10 mm/s). After printing, the conductive silver paste resonant structure is heated by a hot air gun at 50 °C and dried, as shown in Figure 7V. Finally, the upper surface of the metamaterial, as shown in Figure 7VI, is obtained to complete the processing of the metamaterial.

It should be noted that, after optimizing the structural parameters of the metamaterial absorber, we can also first print the upper surface of the metamaterial and then print the lower surface. When printing the resonant structure of the upper surface, it is necessary to use CAD drawing software to draw the resonant unit into a periodic array, to save it as an .stl file, and input it into the slicing software for slicing (that is, printing path planning). Figure 8a shows a schematic diagram of the metamaterial periodic resonance structure after slicing in which there are 25 × 25 units. The whole printing process takes only 10 min. Figure 8b shows a photograph of the upper surface resonant structure after printing; Figure 8c shows a microscopic image of a single unit structure. It can be seen that, although the printed structure has some defects, it is basically consistent with the design.

Next, a fiber-based terahertz time-domain spectroscopy system (THz-TDS) was constructed to measure the spectra of the printed absorber. Figure 9a,b shows a schematic diagram of the fiber-based THz-TDS measurement system; Figure 9c is the physical picture. Figure 9a describes the transmission measurement optical path, while Figure 9b describes the reflective measurement optical path. Because the metamaterial absorber we designed has a conductive ground plane, terahertz waves can hardly be transmitted. Through the actual measurement of the sample transmittance, its transmittance in the entire frequency range is almost 0. Therefore, we set the transmittance to 0, and did not give the measurement results of the transmission spectrum, but only measured the reflection spectrum. The details are described below.

The femtosecond laser enters the optical fiber after passing through the reflector, half-wave plate and lens, and is divided into two beams by the optical fiber splitter. One beam, as the pump light, is incident to the terahertz antenna at the transmitting end to generate terahertz waves, and the other beam, as the detecting light, is incident to the terahertz antenna at the detecting end to detect terahertz pulses. The transmitting antenna radiates a terahertz wave, which is collimated by the off-axis parabolic mirror and then irradiates the sample. After the THz wave is reflected by the sample, it is focused by the second off-axis parabolic mirror, so that the receiving antenna receives the THz signals.

It should be mentioned that the reference plane with reflectivity close to 100% first needs to be selected to obtain the reflection spectrum *R_ref_* of the reference plane. After that, the same position and angle is ensured, the reference plane is changed to the sample to be tested, and the spectrum *R_sam_* of the sample obtained. Then, the absorption spectrum of the sample can be calculated by
(7)$$A=\frac{R_{ref}-R_{sam}}{R_{ref}}$$

The absorption spectrum is shown in Figure 9d. It can be seen that there are two obvious high absorption peaks at 0.215 THz and 0.325 THz, with absorption of 91% and 80%, respectively. However, compared with the simulation results, the absorption spectrum bandwidth measured in the experiment is wider, and there is an absorption peak shift. There are two main reasons for this: First, the conductivity of the resonant structure printed by the electrohydrodynamic jet is lower than that of the conductive silver paste. Although these resonance structures are also composed of conductive silver paste, the uneven distribution of silver paste will occur during the electrohydrodynamic jet process, which affects the conductivity. As the conductivity decreases, the impedance of the resonant structure increases, which increases the absorption bandwidth and moves the resonant peak to the low frequency. Second, the line width of the central hexagonal resonant ring in the metamaterial unit will increase during the process of electrohydrodynamic jet printing. Although the accuracy of the method is very high, the size after curing will be slightly larger than the design due to the use of molten printed materials. As the line width increases, the frequency of the second absorption peak of the absorber will move to a higher frequency. The aforementioned discrepancy notwithstanding, the experiment results agree with the simulation results. This proves the effectiveness of the method of preparing metamaterials by electrohydrodynamic jet printing.

Next, we compared the proposed processing method with other methods, as shown in Table 1. The main comparison features included processing accuracy, system volume, processing cost, and processing time.

It can be seen that, compared with traditional photolithography and etching methods [7], our method has low cost, requires little time and has a smaller volume of the processing system. Compared with the FDM (fused deposition modeling) 3D printing method [47], our method has higher accuracy. Compared with the laser sintering 3D printing method [48], our method has a smaller volume of the processing system and lower cost. Compared with the SLA (stereo lithography appearance) 3D printing method [49], our method has lower cost. Compared with the traditional EHD (electrohydrodynamic) method [50], our method has lower cost and a smaller volume of the processing system.

## 5. Conclusions

In summary, the paper reports a design method for dual-band terahertz metamaterials and a simple, rapid and low-cost method to process metamaterials. The metamaterial design method is based on a sandwich structure, which can achieve polarization insensitive and wide-incidence-angle dual-band absorption. The processing method is based on step-motor-driven electrohydrodynamic jet printing technology. By reforming the commercial or open-source 3D printer, the high-precision stepping motor is used to drive the printing consumables instead of the air pressure, which can accurately control the printing speed and quantity of consumable materials. The whole system has a smaller volume and lower cost. In addition, conductive silver paste is used as the printing material. After optimizing the parameters, the metamaterial structure can be printed directly without complex processes, such as photolithography and etching. The whole printing process only takes a dozen minutes. Finally, a fiber-based terahertz time-domain spectroscopy system was built to test its reflection spectrum. The measured maximum absorption reached 90%. The measured results were in agreement with the simulated results. We believe that the proposed methods will make the design and preparation of metamaterials more flexible, simple, rapid and low-cost. They may help to encourage more innovative applications in the field of metamaterials in the future.

## Figures and Tables

**Figure 1 micromachines-14-00659-f001:**
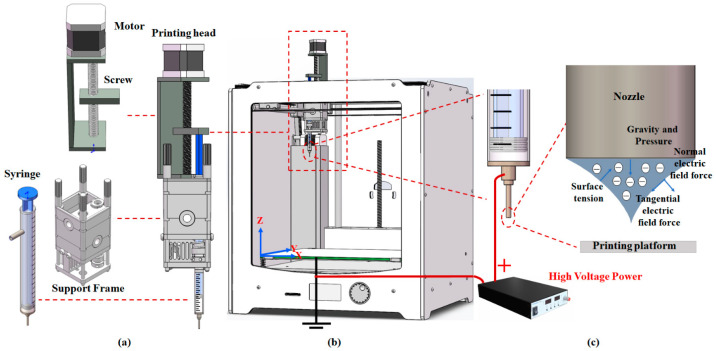
(**a**) E-jet printing head composed of a step drive motor, a screw, a storage syringe and a support frame; (**b**) The print head is installed on an open-source 3D printer through modification; (**c**) Schematic diagram of the Taylor cone generated by applying a high-voltage electric field.

**Figure 2 micromachines-14-00659-f002:**
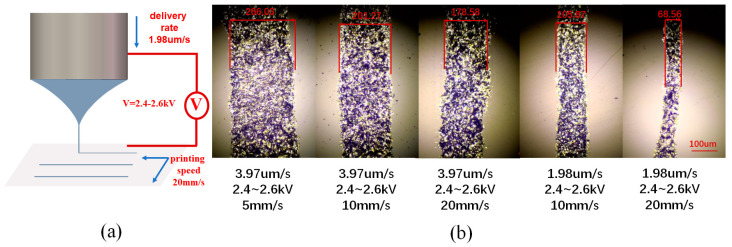
(**a**) Schematic diagram of printing by applied voltage to form a conical jet; (**b**) Five printing effects under different printing parameters.

**Figure 3 micromachines-14-00659-f003:**
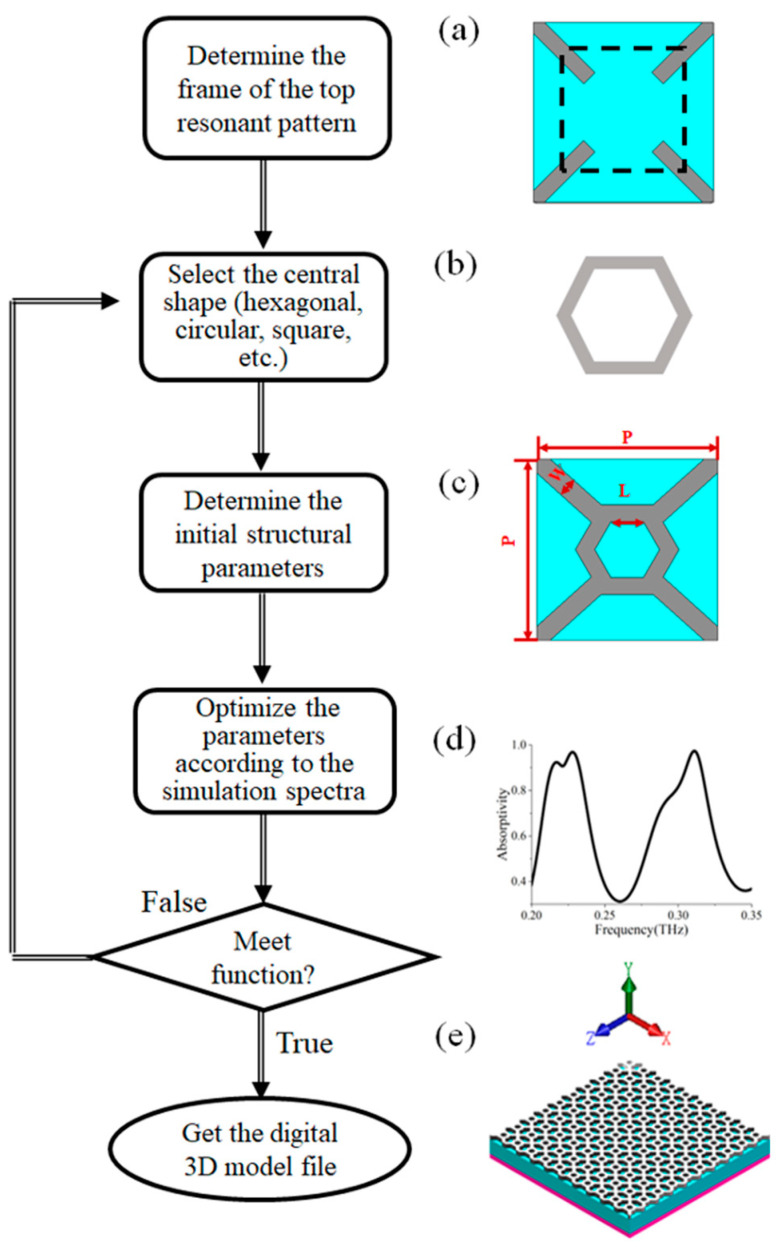
Schematic diagram of the design method. (**a**) The frame which is composed of four diagonal lines at four corners; (**b**) The hexagonal central shape; (**c**) The initial structural parameters of the resonant pattern; (**d**) The absorption spectrum of the absorber; (**e**) The digital 3D model file of the structure.

**Figure 4 micromachines-14-00659-f004:**
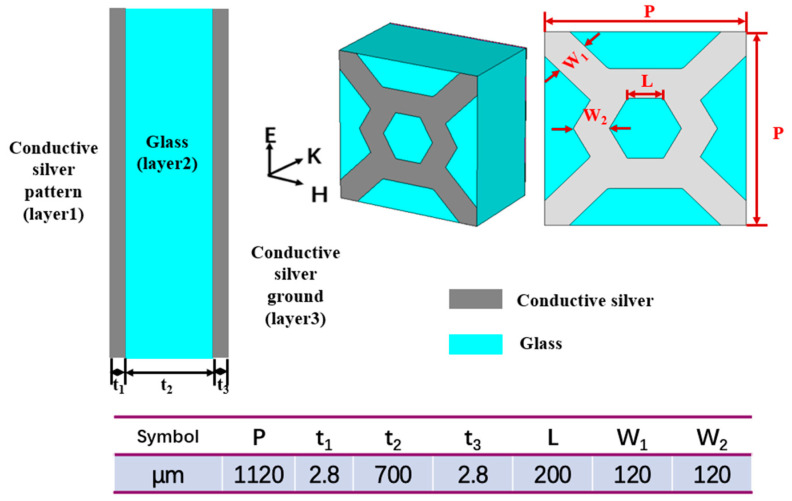
Structure and size diagram of the terahertz metamaterial absorber unit.

**Figure 5 micromachines-14-00659-f005:**
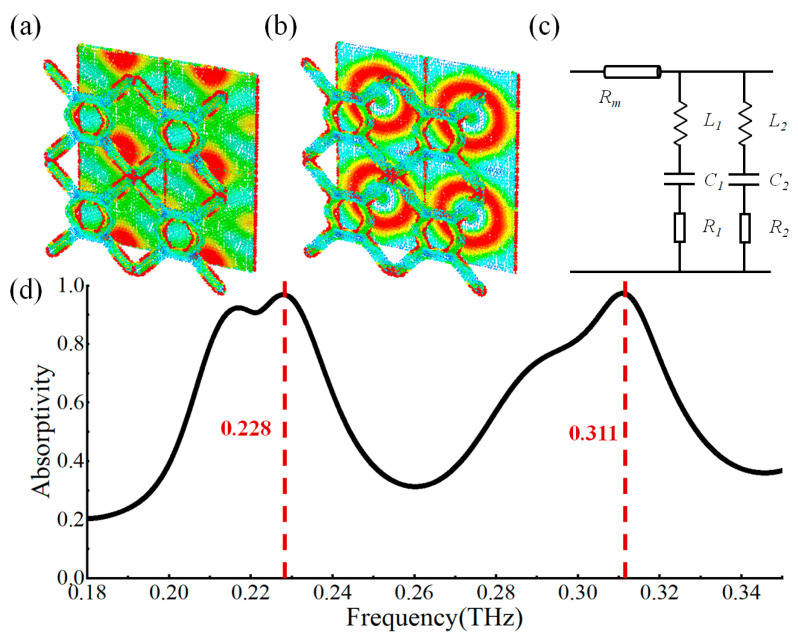
(**a**) Current distribution in the top structure and ground layer of the absorber at 0.228 THz; (**b**) Current distribution in the top structure and ground layer of the absorber at 0.311 THz; (**c**) Transmission line model of the structure; (**d**) The simulated absorptivity spectrum.

**Figure 6 micromachines-14-00659-f006:**
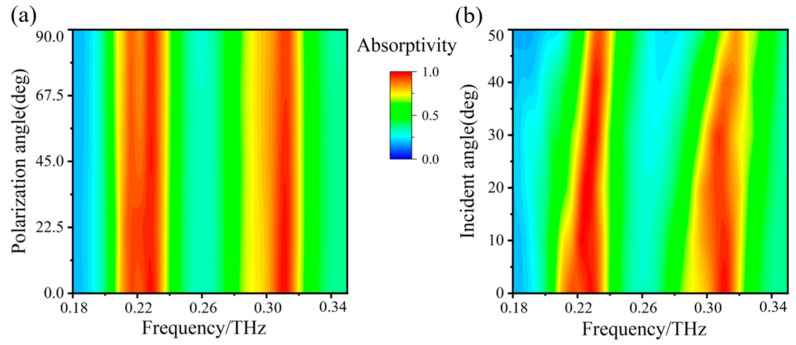
Effect of (**a**) polarization angle and (**b**) incident angle on absorptivity.

**Figure 7 micromachines-14-00659-f007:**
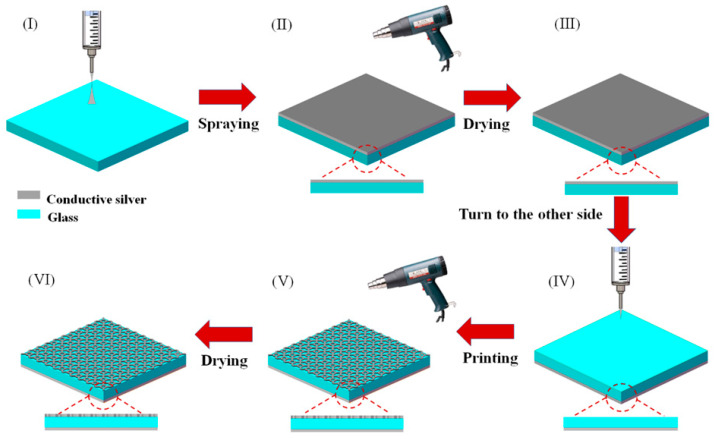
Flow of preparing the terahertz metamaterial absorber by electro-hydrodynamic jet printing. (**I**) Spray the conductive layer on the surface of the glass; (**II**) Dry the sprayed conductive silver paste layer; (**III**) Obtain the ground layer of the metamaterial; (**IV**) Print the resonance structure on the upper surface of the glass; (**V**) Dry the printed the resonance structure; (**VI**) Complete the top layer of the metamaterial.

**Figure 8 micromachines-14-00659-f008:**
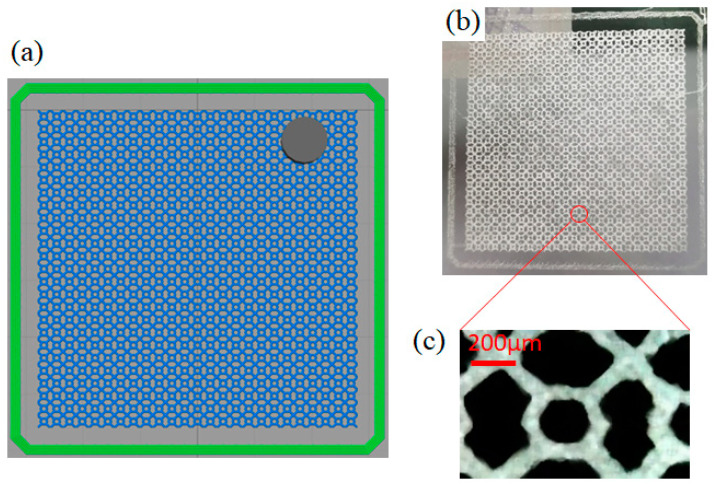
(**a**) Schematic diagram of metamaterial periodic resonance structure after slicing; (**b**) Photo of the upper surface resonant structure; (**c**) Microscopic image of a single unit structure.

**Figure 9 micromachines-14-00659-f009:**
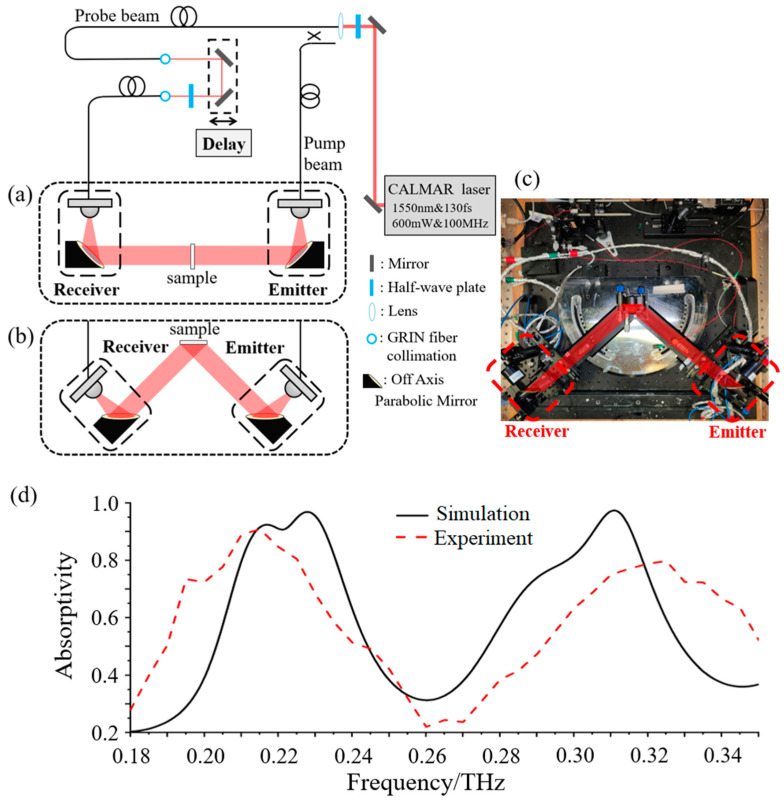
(**a**) Schematic diagram of the transmission spectrum measurement; (**b**) Schematic diagram of reflection spectrum measurement; (**c**) Physical image of the transmission and reflection integrated THz-TDS system; (**d**) Simulation and measurement results of the absorption spectrum.

**Table 1 micromachines-14-00659-t001:** Comparison of our work with other preparation methods.

Method	Accuracy	Volume	Cost	Time
Photolithography [7]	High	Large	High	Long
FDM [47]	Low	Small	Low	Short
Laser sintering [48]	High	Large	High	Short
SLA [49]	High	Small	High	Short
Traditional EHD [50]	High	Large	High	Short
Our work	High	Small	Low	Short

## Data Availability

Data underlying the results presented in this paper are not publicly available at this time but may be obtained from the authors upon reasonable request.

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
