# Peer review of "Design and Print Terahertz Metamaterials Based on Electrohydrodynamic Jet"

_micromachines, 2023, doi:10.3390/mi14030659_

Round 1

Reviewer 1 Report

In this paper, a design method of dual-band terahertz metamaterials and a simple, rapid, low-cost  metamaterial preparation scheme based on step-motor driven electrohydrodynamic jet technology are proposed. I think, paper is interesting, I would propose some changes as follows:

1.      It is not clear from Eq. 1 if the used permittivity is the effective one obtained after homogenization of the metamaterial has been performed.

2.      Absorptivity is plotted in Fig. 5 in this relation its physical origin should be discussed.

3.      Authors should stress novelty of their work in comparison with others.

4.      Authors are missing some recent articles in the field such as Investigation of Hyperbolic Metamaterials, etc.

Author Response

We gratefully appreciate the precious time that the referee have spent on our manuscript. The valuable comments are significant to promote the quality of our research. Below, we detail our responses for each comment. Our list of responses is in the same order posed by the referees. All changes have been indicated the locations. In addition, the changes have been highlighted with a red color in the revised manuscript. Please see the attachment “respons_letter_Reviewer1.pdf”.

Reviewer 2 Report

Comments on the manuscript

In this study, the authors proposed a simple, cost-efficient, and time-saving fabrication method for creating THz metasurfaces using electrohydrodynamic jet printing. The technique's success centers on controlling the feed flow rate and externally applied electric field. The authors validated their approach through experiments, which showed dual-band (almost) perfect absorptions in the THz range, in line with simulation predictions.

The overall content of the manuscript is intriguing, and the experimental outcomes provide a valuable contribution to the field of metasurface fabrication. Furthermore, the manuscript is methodologically robust, with well-substantiated claims and conclusions. Consequently, this manuscript aligns with the scope of Micromachines. I recommend that this work be published with minor revisions, which I have outlined below

1.       I am primarily concerned about the purity of the conductive silver paste. Does the mixture have a similar permittivity to real silver that is prepared by standard thermal/E-beam evaporation? I am asking this question because of the overall absorption spectrum. The fabricated metasurfaces seem to have strong absorption even at non-resonance wavelengths, as shown in Figure 9. Can you comment on which aspect of the metasurface design is causing this loss? This step is crucial as it is closely related to the technical innovations and scientific impact of the manuscript.

2.       “Through continuous improvement, metamaterial absorbers have developed from microwave band [10,11] to terahertz [12,13], infrared [14,15], and even visible light band [16,17].” The statement presents an interesting perspective by shedding light on recent progress across different wavelength ranges. However, it would be beneficial if the authors could elaborate on this aspect by incorporating discussions on cutting-edge developments in mid-IR metamaterials absorbers, especially regarding advanced thermal management. The works of Yuri Kivshar et al. [Nano Letters, 21(20), 8917-8923] and Willie J. Padilla et al. [Nano Lett. 2021, 21, 9, 4106–4114], for example, go into this issue with intricate details and may bring useful insights.

3.       How do your results relate to the transmission measurement scheme shown in Figure 9a?

Author Response

We gratefully appreciate the precious time that the referee have spent on our manuscript. The valuable comments are significant to promote the quality of our research. Below, we detail our responses for each comment. Our list of responses is in the same order posed by the referees. All changes have been indicated the locations. In addition, the changes have been highlighted with a red color in the revised manuscript.Please see the attachment "respons_letter_Reviewer2.pdf".
